# Effect of Prepartum Dietary Energy Level on Production and Reproduction in Nili Ravi Buffaloes

**DOI:** 10.3390/ani12131683

**Published:** 2022-06-30

**Authors:** Muhammad Binyameen, Muhammad Irfan ur Rehman Khan, Muhmmad Naveed Ul Haque, Burhan E. Azam, Akke Kok, Ariette T. M. Van Knegsel, Muhammad Zahid Tahir

**Affiliations:** 1Buffalo Research Institute, Pattoki 55300, Pakistan; drbinyameen@yahoo.com (M.B.); dr.burhanrana77@gmail.com (B.E.A.); 2Department of Theriogenology, University of Veterinary and Animal Sciences, Lahore 54000, Pakistan; irfan.khan@uvas.edu.pk; 3Department of Animal Nutrition, University of Veterinary and Animal Sciences, Lahore 54000, Pakistan; muhammad.naveed@uvas.edu.pk; 4Adaptation Physiology Group, Wageningen University and Research, 6700 AH Wageningen, The Netherlands; akke.kok@wur.nl (A.K.); ariette.vanknegsel@wur.nl (A.T.M.V.K.)

**Keywords:** transition period, milk composition, buffalo, postpartum, nutrition

## Abstract

**Simple Summary:**

The objective was to evaluate the effect of prepartum dietary energy level on performance of Nili Ravi buffaloes. Buffaloes were fed low, medium or high energy diet during 63 days prepartum and same lactation diet during 14 weeks postpartum. Buffaloes fed the high energy diet had better performance compared with buffaloes fed the medium or low energy diet. While, prepartum feeding of low energy diet adversely affected buffalo performance.

**Abstract:**

The objective of this study was to evaluate the effect of prepartum dietary energy level on postpartum production and reproduction in Nili Ravi buffaloes (*n* = 21). The buffaloes were offered low energy (LE: 1.31 Mcal/kg DM NE_L_ (net energy for lactation)), medium energy (ME: 1.42 Mcal/kg DM NE_L_) or high energy (HE: 1.54 Mcal/kg DM NE_L_) diet for 63 days prepartum, and received the same lactation diet (LD: 1.22 Mcal/kg DM NE_L_) during 14 weeks postpartum. The effects of dietary energy level and week were analyzed with Proc GLIMMIX model. Dry matter intake (DMI) was lower in buffaloes fed the LE diet compared with buffaloes fed the ME or HE diet. Calf birth weight (CBW) was higher in buffaloes fed the HE diet compared with buffaloes fed the ME or LE diet. Milk production was similar in buffaloes fed the HE, ME or LE diet within 14 weeks postpartum and throughout the lactation. Milk fat was higher in buffaloes fed the LE diet compared with buffaloes fed the ME or HE diet. Milk protein and lactose yields was high in buffaloes fed the HE diet compared with buffaloes fed the ME or LE diet. Body condition score was high in HE and was affected by diet × week interactions during pre and postpartum period. The concentrations of β-hydroxybutyrate (BHBA) and triglycerides in serum was lowest in buffaloes fed the HE diet compared with the buffaloes fed the ME or LE diet. The buffaloes fed the HE diet had early uterine involution (UI), first estrus, short dry days, and calving interval (CI) compared with buffaloes fed the ME or LE diet. None of buffaloes fed the LE diet exhibited estrus during the first 14 weeks postpartum compared with buffaloes fed the ME or HE diet. In conclusion, prepartum feeding of high energy diet can be helpful in improving the postpartum productive and reproductive performance in Nili Ravi buffaloes.

## 1. Introduction

Buffaloes (*Bubalis bubalis*) have been reported in 67 countries with 202 million heads across the world [1]. Buffaloes are divided mainly into river and swamp types [2]. River buffaloes are mostly used for milk and meat production, while swamp buffaloes are used for drought purposes [3]. Among river buffaloes, Nili Ravi is the principal milk producing breed [4] selected for dairy production [5]. The performance of buffalo is controlled by various factors, such as genetics [6], management [7], nutrition [8], and, most importantly, reproduction [9].

The reproductive performance of buffaloes is directly affected by animal associated anomalies, such as postpartum ovarian activity [10], uterine involution [11], timing of ovulation [12], and insemination [13]. Other contributing factors hampering buffalo reproductive performance are: parity [14], seasonality [15], suckling [16], body condition score (BCS) [17], and nutritional status [18]. The poor reproductive performance, ultimately, results in a prolonged calving interval, decreased milk yield, culling of infertile animals, and additional feed costs for non-productive animals [19,20]. While prolonged calving interval hampers the buffalo farming profitability, it can be reduced through prepartum nutritional interventions.

The role of nutritional strategies during the transition period has been well-established for improved health, production, and fertility in dairy cows [21]. In buffaloes, there is a scarcity of knowledge regarding their nutritional needs as per the changing physiological status [22]. Consequently, extrapolation from NRC (2001) standards of cattle feeding has been suggested for buffaloes [23]. However, in routine practice, buffaloes are fed with seasonal fodder, such as sorghum or maize during the summer season and forage grasses during the winter season [24]. Along with seasonal fodder, buffaloes remain on crop residues, such as wheat and rice straws without concentrate supplementation, that hampers the productive and reproductive performance in buffaloes [18]. Therefore, it can be hypothesized that feeding of appropriate dietary energy level during the dry period is critical for sustainable buffalo farming. As a result, the first objective of this study was to evaluate the effect of prepartum dietary energy level on DMI, calf weights, milk production, milk composition, body weight, BCS, and blood metabolites. The second objective was to determine the effect of prepartum dietary energy level on uterine involution, ovarian resumption, onset of estrus, days open, conception rate, pregnancy rate, and calving interval.

## 2. Materials and Methods

### 2.1. Experimental Design, Diets and Housing

The experiment was conducted at Livestock Experiment station (LES) Bhunikey, Pattoki District Kasur, Punjab Pakistan (31°1′30.0324″ N and 73°50′52.3608″ E) throughout the month of October 2019–March 2020. The study comprised of 21 adult Nili Ravi buffaloes of 2–5 parity and 2.5–4 BCS (1–5) scale, as described by Edmonson et al. [25]. The diets were formulated as per the guidelines of Cornell Pen Minor 3.0.10 (CPM) software (Cornell University, Ithaca, NY, USA). The composition and nutritional analyses of prepartum and postpartum diets are presented in Table 1. The animals were offered low energy (LE: 1.31 Mcal/kg DM NE_L_ (net energy for lactation), medium energy (ME: 1.42 Mcal/kg DM NE_L_) or high energy (HE: 1.54 Mcal/kg DM NE_L_) diet during 63 days prepartum and maintained at lactation diet (LD: 1.22 Mcal/kg DM NE_L_) during 14 weeks postpartum. The animals were fed individually in a separate tie-stall barn at 10:00 a.m. once daily during the prepartum period. During the postpartum period, animals were fed in groups. All animals had free access to fresh water throughout the study. All calves were maintained on similar diet (2 kg of hay, 300-g of calf starter and milk as per body weight allowance) from birth to weaning (14 weeks).

### 2.2. Sample Collection and Analysis

Diets offered as well as individual orts were weighed daily. Samples of each feedstuff were collected weekly to evaluate the DM and for further laboratory analysis. These samples were analyzed for DM, CP, NDF, ADF, ether extract, and ash, in accordance with the official methods of [26]. The calf birth weight (CBW) was recorded within 30 min after birth and body weights every week until 14 weeks. The milk production was recorded daily at 0600 and 18:00 p.m. until the end of lactation, while milk composition was analyzed weekly for fat, protein, and lactose for 12 weeks (Lacto Scan Milk Analyzer, S-60, V-62, Burgas, Bulgaria). The calves were separated from the dams at the time of birth, but dams refused to let down the milk. Therefore, the milk production was recorded by calculating milk from one side of udder alternatively and was multiplied by two. The complete milk production of every animal was also confirmed by complete milking on a fixed day weekly by tying the calf before the dam every week without any suckling. The dam’s body weight (BW) was recorded after milking and before feed distribution on fortnightly basis. The dam’s body condition score (BCS) was recorded by a single person during evening milking fortnightly. Blood samples were taken at −17, −10, −3, +3, +10, +17 days from jugular vein prior to feeding and 4 h after milking using vacutainers without any additives. Serum samples were collected after centrifugation at 2000× *g* for 15 min and samples were stored at −64 °C pending analysis. Serum samples were analyzed through enzymatic diagnostic commercial kits (Randox Laboratories, Ltd.) for the concentration of glucose (GL2623), blood urea nitrogen (BUN) (UR, 107), triglycerides (TR, 210), and β-hydroxybutyrate (BHBA) (RB1007) using biochemical analyzer (Rx, Monza, Randox Labortories, Ltd.) method, as described by [27,28].

The first follicle formation (FFF), resumption of ovarian activity (OR), and appearance of first corpus luteum postpartum (FCL) were recorded from 14 days postpartum through transrectal probe (Honda 7400, Tokyo, Japan) and were repeated precisely every 3 days. The biometry of reproductive tract started 14 days postpartum and was repeated every 3 days until complete uterine involution (UI). The criteria for UI were set in accordance with [29]. The diameter and length of uterine horns were recorded at each palpation. In addition, uterine involution was considered complete until no further reduction in size of horn was observed for two successive recordings. Estrus was detected after morning and evening milking with penile deviated teaser bull 40 to 100 days postpartum. Estrus was confirmed by measuring the largest follicle on the ovary and insemination with frozen semen was performed by a single technician 24 h after standing heat [30]. The presence of a large follicle at estrus and its absence after 24 h confirmed ovulation. Non-return rate (NRR), conception rate (CR), and pregnancy rate (PR) were confirmed between 20–24, 32–35, and 58–60 days after artificial insemination (AI), respectively.

### 2.3. Statistical Analysis

Prior to statistical analysis, data collected on daily basis were condensed into weekly means. The prepartum (−9 to 0 weeks) and postpartum (1 to 14 weeks) data were analyzed separately. The repeated measure analysis was performed for variables measured over time, such as DMI, BW, BCS, milk production, and composition. Repeated measure data were analyzed through Proc GLIMMIX of SAS University Edition [31]. The treatments, weeks, treatment × week interactions were fixed effect, while the animals were random effect. The Satterthwaite option was used to calculate the degrees of freedom. The GLIMMIX procedure of SAS was used without repeated measures for weeks relative to calving, and interactions were removed for variables not measured over time from the model, such as total milk yield, calf birth weights, uterine involution, first estrus expression, open days, and calving interval. Level of significance was set (*p*-value < 0.05) and tendency was set at *p* > 0.05 and *p* < 0.10. Reproductive variables (NRR, CR, and PR) were analyzed by using the chi-square test.

## 3. Results

### 3.1. Dry Matter Intake

Dry matter intake (DMI) in the 9 weeks prepartum was affected by diet (12.2 vs. 13.4 vs. 13.8 ± 0.23 for the LE, ME or HE diet, respectively) (*p* < 0.01) and tended to be affected by week (*p* = 0.06, Figure 1).

### 3.2. Milk Production, Milk Composition and Lactation Length

There was no effect of prepartum diet on daily milk production during the first 14 weeks (Figure 2), and throughout the lactation (*p* > 0.10, Table 2). Milk production was affected by week during the first 14 weeks and throughout the lactation (*p* < 0.01). The milk fat, protein, and lactose % were affected by week (*p* < 0.01) during the first 12 weeks postpartum. Milk fat yield was affected by diet × week interactions (*p* < 0.01). Milk lactose yield tended to be affected by dietary treatments (*p* = 0.09). Milk protein and lactose yields were higher in the HE diet fed buffaloes compared with the LE diet fed buffaloes (362 and 481 for HE diet vs. 268 and 355 g/day for LE diet). Total milk production, number of days in milk, and subsequent dry period length were also similar (*p* > 0.10).

### 3.3. BCS and Body Weight

Body condition score was affected by diet × week interactions (Table 3) during pre and postpartum period (*p* < 0.05, Figure 3). Body weight of the dams was affected by week (*p* < 0.01), but not by diet (*p* > 0.10) during pre and postpartum period (Figure 4). Calf birth weight was higher (*p* = 0.01) for buffaloes fed the HE diet compared with buffaloes fed the LE diet (*p* = 0.01), and tended to be higher for buffaloes fed the HE diet compared with buffaloes fed the ME diet (*p* = 0.07; 41 vs. 35 vs. 32 ± 1.80 kg for HE vs. ME vs. LE diet). However, body weight of the calves at 14 weeks postpartum was not affected by diet (Figure 5).

### 3.4. Serum Metabolites

Prepartum, serum metabolites were not affected by diet (Table 4). Postpartum, serum glucose tended to be affected (*p* = 0.10) and triglycerides were lower (*p* < 0.05) for buffaloes fed the HE diet compared with buffaloes fed the ME or LE diet (Figure 6). The serum concentration of BHBA was affected by week × diets (*p* < 0.05) and it was lower in HE diet compared with LE or ME diet during postpartum (Figure 6d).

### 3.5. Reproductive Performance

Prepartum diets did not affect the timing of first follicle formation, timing of resumption of ovarian cyclicity, and timing of first CL formation during postpartum period (Table 5). Uterine involution was earlier (*p* < 0.05) for buffaloes fed the HE diet compared with buffaloes fed the ME or LE diet. The first estrus was earlier (*p* < 0.01) and days open tended (*p* = 0.06) to be earlier for buffaloes fed the HE diet compared with buffaloes fed the ME or LE diet. In addition, number of inseminations per pregnancy was not affected by diet. Buffaloes fed the LE diet did not express any estrus signs during the first 14 weeks postpartum. Of this group, two animals failed to conceive and were culled due to sever emaciation at the end of lactation. Calving interval was shorter (*p* < 0.01) for buffaloes fed the HE diet compared with buffaloes fed the ME or LE diet.

## 4. Discussion

The objective of this study was to evaluate the effect of prepartum dietary energy level on production and reproduction of Nili Ravi buffaloes. All experimental buffaloes remained on lactation diet (LD) for 14 weeks postpartum, as shown in Table 1. After 14 weeks, animals remained on homogeneous conventional diet until next calving. The HE diet had a greater DMI prepartum compared with buffaloes fed the LE diet. Greater DMI during prepartum was probably due to the more concentrate supplementation in the diet [8,32]. The results are in contrast to [33], who reported that different energy levels had similar DMI in buffaloes. The contradiction might be due to the difference in physiological stages, as they reported DMI in lactating buffaloes. The greater DMI in HE diet increased the body energy reserves and helped in reducing BCS changes of animals during transition. Our findings are similar to [18,32], who reported that greater prepartum energy intake reduced the BCS changes in buffaloes. The low DMI intake in LE diet might be due to the increased quantity of wheat straw that decreased palatability [34,35]. The low DMI during prepartum possibly lowered the energy intake, body reserves, and exerted negative influence on reproductive performance. Our findings are in line with [32,36], who reported that lower dry matter intake impaired the reproductive performance of buffalo heifers.

The daily milk production during first 14 weeks postpartum and throughout lactation was similar for buffaloes fed the LE, ME or HE diet during the prepartum period. Total milk production at the end of lactation was also similar for buffaloes fed the LE, ME or HE diet. However, milk production was numerically higher in buffaloes fed the HE diet compared with buffaloes fed the LE diet during first 14 weeks in lactation (9.7 vs. 7.6 kg/d for HE vs. LE diet, Figure 2), throughout the lactation (8.7 vs. 6.6 kg/d for HE vs. LE) and as a total production per lactation (2198 vs. 1642 kg/d for HE vs. LE). The current findings are similar to [37], who reported no effect of prepartum diet on milk yield in buffaloes. Our findings are also similar to [38,39], who reported that prepartum diet manipulation did not exert any influence on milk production during early lactation in cows. These findings are in contrast to [32,40,41], who reported that prepartum higher energy feeding increased milk production in buffaloes. It might be due to composition (green fodder and less quantity of concentrate) and duration of prepartum dietary treatment (75 and 90 days). A possible reason for greater numerical milk production for buffaloes fed the HE diet could be better energy reserves and greater glucose utilization for milk synthesis. Serum glucose concentration was lower for buffaloes fed the HE diet during postpartum period compared with buffaloes fed the ME or LE diet. Our findings are supported by [42], who reported negative correlation of blood glucose with milk production in dairy cows. Milk lactose contents were higher in the HE diet fed buffaloes, which indicates that high lactose contents were responsible for more milk production. In addition, low milk fat in HE group indicates that minimum fat reserves have been mobilized by the HE diet fed buffaloes compared with the ME or LE diet fed buffaloes. These findings are supported by [43], who reported that high energy availability promoted milk production with low fat mobilization in buffaloes. The results are in contrast to [44], who reported greater postpartum BCS loss in high yielding buffaloes. It might be due to short dry period in their study and the fact that animals were unable to store their energy reserves for postpartum period. High fat contents have been reported in LE diet and were due to more mobilization of body reserves, as reported in dairy cows by [45,46].

In the current study, BCS change was lowest for buffaloes fed the HE diet before, during, and after parturition compared with buffaloes fed ME or LE diet. The lowest concentration of triglycerides and BHBA in serum of HE diet reflected minimum BCS change during the postpartum period. These findings are similar to [32,40], who reported high BCS in buffaloes that were fed higher energy diet compared with moderate energy during prepartum period. It can be hypothesized that the greater DMI during the abovementioned prepartum period exerted a carry-over effect on postpartum DMI. Consequently, BCS change remained at minimum for buffaloes fed the HE diet. The relatively low BCS change in HE diet compared with ME or LE diet could be due to more DMI, more developed ruminal papillae, and more adoption of rumen for higher starch content in postpartum diet [47]. Our findings are similar to [18], who reported inactive ovaries in low BCS buffaloes. Our findings are in line with [48], who reported no difference in milk yield between low or high BCS buffaloes at calving. While maximum BCS loss from LE diet did not exert any effect to increase the milk production. Moreover, these findings are in contrast to [49], who reported that cows with maximum BCS loss had improved milk production.

Body weights did not change during pre and postpartum period for the LE, ME or HE diets fed buffaloes. These results are similar to [37], who reported no change in body weight due to prepartum concentrate supplementation during pre and postpartum period in buffaloes. The birth weights of the calves were higher for buffaloes fed the HE diet compared with ME or LE diet. During the last trimester of pregnancy, fetus growth is high and nutrient demand of the calf reaches a maximum before calving [50]. In addition, HE diet improved the birth weights in buffaloes [40,41,51]. These results are in contrast to [37], who reported no effect of prepartum concentrate supplementation on birth weights of buffalo calves. Differences might be due to dietary treatments and they offered diet during close up period of 3 weeks prepartum, while in the current study buffaloes were fed 9 weeks prepartum. The authors of [52] postulated that the decrease in DMI towards parturition in dairy cows might be due to the increasing size of fetus, which restricts rumen volume. Interestingly, and in contrary to the general perception, it seems that the increasing weight of fetus towards the end of gestation did not decrease DMI in HE diet. These findings are in line with [53], who reported improved birth weights after improved feeding in swamp buffaloes. Weight of calves at 14 weeks postpartum was not affected by prepartum dietary treatments. Similar findings have been reported by [39,40] in buffalo calves.

The concentration of glucose in serum remained unchanged during pre and postpartum period. It indicated a smooth rate of gluconeogenesis before and after parturition and similar findings have been reported by [54,55] in buffaloes. The results are in contrast to [56], who reported higher blood glucose in herbal supplemented buffaloes compared with control. Differences might be due to feeding of *Asparagus racemosus* in their study. Similarly, BUN concentration remained unaltered during pre and postpartum period. These results are in contrast to [57], who reported high BUN during postpartum period in buffaloes. Differences might be due to more concentrate feeding during postpartum period in their study. The concentration of triglyceride was low in the HE diet fed buffaloes compared with the ME or LE diet fed buffaloes. These results are in line with [56], who reported different levels of triglyceride during pre and postpartum period in buffaloes. While these results are contrary to [58], who reported no change in triglyceride levels during pre and postpartum period. It might be due to differences in dietary treatment and duration of treatment (protein and fat feeding; 60 and 85 days). The BHBA concentration was lower in HE diet fed buffaloes compared with LE diet fed buffaloes. In addition, the results are supported by [59], who reported low concentration of BHBA during postpartum period in nutritional supplemented buffaloes.

Prepartum feeding of high energy diet positively affected the reproductive variables of buffaloes. The uterine involution, first estrus expression, and days open were early in buffaloes fed the HE diet compared with buffaloes fed ME or LE diet. These findings are supported by [60,61] and they showed a strong correlation between uterine involution and first ovulation in buffaloes. High milk production has no negative effect on buffalo reproductive parameters and similar results have been reported by [62] in cows. In our study, apparently more milk production of HE diet might have exerted a positive influence on uterine involution, but no effect [63,64] has been reported in buffaloes. Uterine involution was 43.5 days for HE diet compared with [59,64,65], who reported uterine involution between 33 to 41 days in buffaloes. The difference might be due to the difference in breed, use of oxytocin, and interval (after 7 days) for measurement of uterine involution in their study. Timing of uterine involution in buffaloes varies among buffaloes ranging from 15 until 74 days maximum in buffaloes [61,66]. The results are in contrast to [67], who reported no effect of prepartum nutrition on uterine involution. It might be due to the duration of reproductive organ measurement as they were recorded every 4 days compared with our study after 3 days. The results are in contrary to [18,67], who reported no or negative impact of prepartum diet on postpartum reproductive performance in buffaloes. These results are in line with [68], who reported a strong correlation of uterine involution with pregnancy. First CL formation was at day 23 in HE diet and [69] also reported FCL at day 23.8 in Nili Ravi buffaloes. The results are also supported by [70], who reported that milk production did not exert any influence on resumption of follicular activity in buffaloes. In addition, improved feeding during prepartum period is responsible for subsequent fertility [41,51]. Our findings are in contrast to [37], who reported high fertility with low and medium concentrate compared with high concentrate supplementation during prepartum in primiparous Nili Ravi buffaloes. Differences might be due to parity (primiparous) and duration of dietary treatment as they started feeding at 150 days of pregnancy. The BCS of HE diet remained higher during pre and postpartum. These findings are supported by [62], who reported that pre and postpartum BCS affect the reproductive performance. The low BCS from LE diet fed buffaloes exerted its effect on reduced fertility and similar findings have been reported by [10,17,18] in buffaloes.

First estrus expression, days open, and most importantly calving interval reduced in buffaloes fed the HE diet compared with buffaloes fed the ME or LE diet. These findings are supported by [53,58,59], who reported reduced open days in peripartum nutritional supplemented buffaloes. The current study reduced the calving interval through HE diet. In our study, calving intervals were 433, 485 or 567 days for buffaloes fed the HE, ME or LE diet, while [71] reported 481 days in Nili Ravi buffaloes from the data collected between 2010 to 2018 at the same farm. While contrary to these findings, the authors of [72] reported 740 days of prolonged calving interval for Murrah cross buffaloes. The results are supported by [18,22,58], who reported that insufficient supply of nutrients is responsible for poor productive and reproductive performance of buffaloes. Our findings are supported by [73], who reported that improved feeding during prepartum period can improve the performance of buffaloes. Although the HE diet exerted a positive influence on buffalo production and reproduction, further studies on postpartum feeding of different energy levels will be interesting to improve the performance of buffaloes.

## 5. Conclusions

Prepartum feeding of high energy diet had greater prepartum DMI, calf birth weights, reduced BCS change, low postpartum triglycerides and BHBA concentrations, earlier timing of uterine involution, first estrus expression, less days open, and short calving interval compared with medium or low energy diet.

## Figures and Tables

**Figure 1 animals-12-01683-f001:**
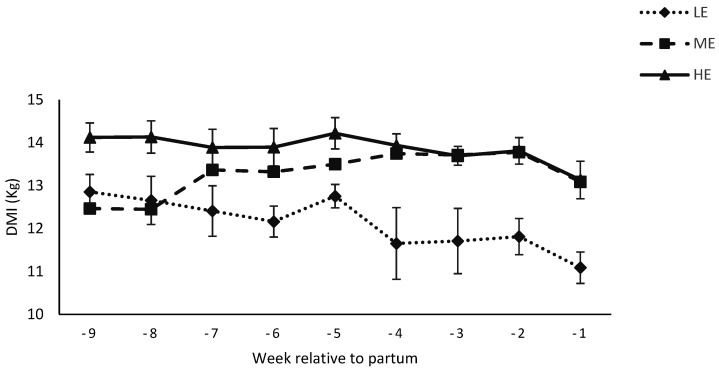
Dry matter intake (DMI; kg/d) of Nili Ravi buffaloes fed a low energy (LE), medium energy (ME) or high energy (HE) level during 9 weeks prepartum. Low, medium, and high energy levels were NE_L_: 1.30, 1.42, and 1.53 Mcal/kg, respectively, on DM basis in dietary treatments. Values represent LS means ± SEM.

**Figure 2 animals-12-01683-f002:**
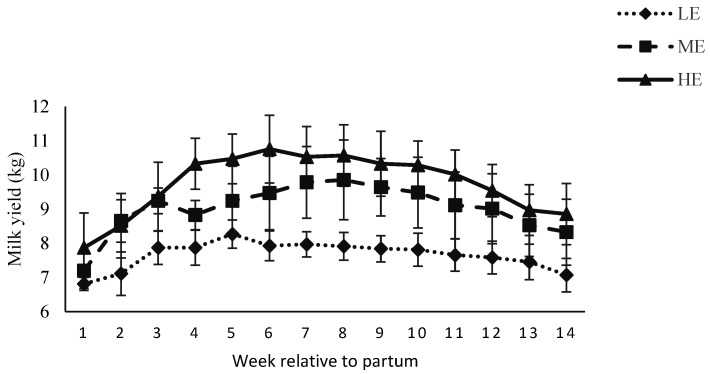
Milk yield (kg/d) of Nili Ravi buffaloes fed a low energy (LE), medium energy (ME) or high energy (HE) level during 9 weeks prepartum. Low, medium, and high energy levels were NE_L_: 1.30, 1.42, and 1.53 Mcal/kg, respectively, on DM basis in dietary treatments. Values represent LS means ± SEM.

**Figure 3 animals-12-01683-f003:**
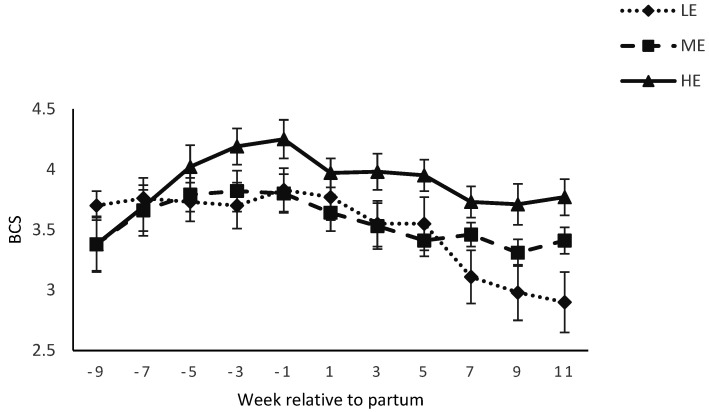
Body condition score (BCS; 1–5) of buffaloes fed a low energy (LE), medium energy (ME) or high energy (HE) level during 9 weeks prepartum. Low, medium, and high energy levels were NE_L_: 1.30, 1.42, and 1.53 Mcal/kg, respectively, on DM basis in dietary treatments. Values represent LS means ± SEM.

**Figure 4 animals-12-01683-f004:**
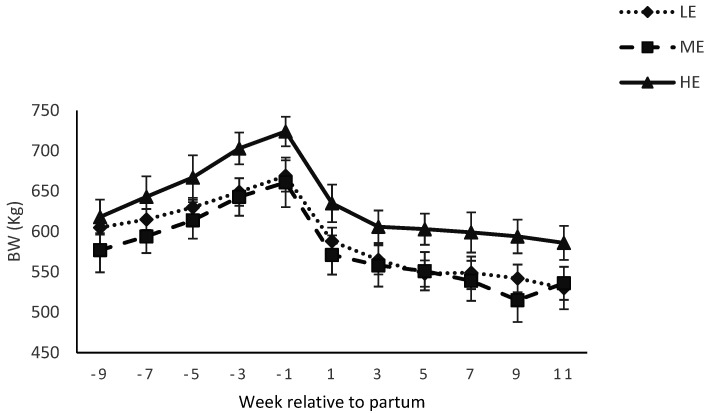
Body weight (BW; kg) of Nili Ravi buffaloes fed a low energy (LE), medium energy (ME) or high energy (HE) level during 9 weeks prepartum. Low, medium, and high energy levels were NE_L_: 1.30, 1.42, and 1.53 Mcal/kg, respectively, on DM basis in dietary treatments. Values represent LS means ± SEM.

**Figure 5 animals-12-01683-f005:**
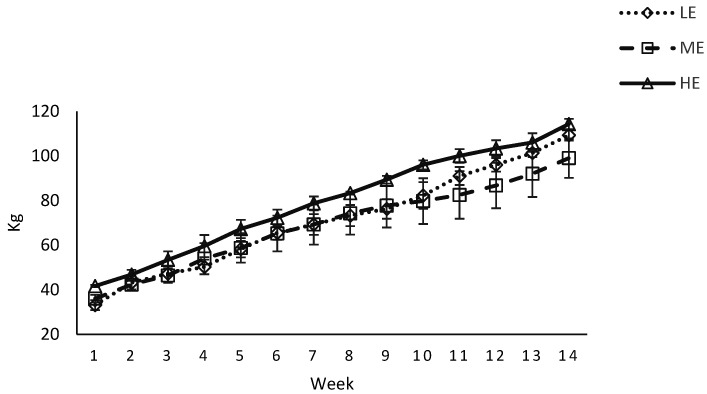
Body weight (BW; kg) of Nili Ravi buffalo calves fed a low energy (LE), medium energy (ME) or high energy (HE) level during 9 weeks prepartum. Low, medium, and high energy levels were NE_L_: 1.30, 1.42, and 1.53 Mcal/kg, respectively, on DM basis in dietary treatments. Values represent LS means ± SEM.

**Figure 6 animals-12-01683-f006:**
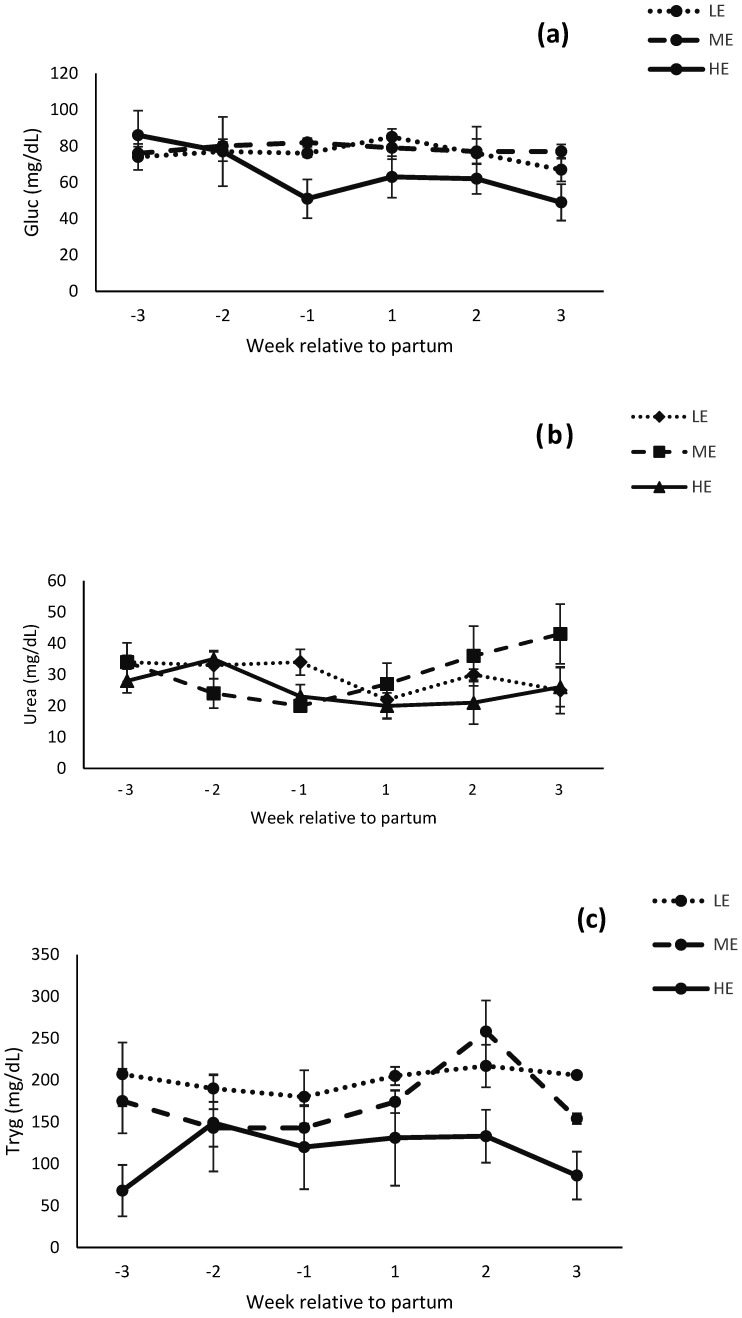
Serum metabolites glucose (**a**), blood urea nitrogen (**b**), triglycerides (**c**), and beta hydroxybutyrate (**d**) (mg/dL) of Nili Ravi buffaloes fed a low energy (LE), medium energy (ME) or high energy (HE) level during 9 weeks prepartum. Low, medium, and high energy levels were NE_L_: 1.30, 1.42, and 1.53 Mcal/kg, respectively, on DM basis in dietary treatments. Values represent LS means ± SEM.

**Table 1 animals-12-01683-t001:** Ingredient and nutrient composition of the diets.

	Dietary Treatments ^1^			
Items	LE	ME	HE	LD
Ingredients, % of DM				
Corn silage	41.90	48.3	49.4	--
Oat Fodder	--	--	--	71.3
Wheat straw	33.6	24.2	14.8	13.1
Corn Grain	1.20	6.50	14.4	2.08
Molasses	0.49	3.35	6.09	1.69
Wheat bran	1.18	1.17	1.14	5.69
Soybean Husk	7.26	2.39	1.17	--
Soybean Meal	5.98	5.32	4.33	--
Canola meal	--	--	--	1.29
Rapeseed meal	--	--	--	1.31
Corn gluten meal 30%	7.78	8.27	8.09	3.21
Di-calcium phosphate	0.20	0.20	0.19	--
Sodium bicarbonate	--	--	--	0.09
Mineral premix	0.40	0.39	0.38	0.27
Vitamin premix	--	--	--	0.09
Nutrient Composition, % of DM				
DM ^2^ (%)	54.5	51.1	50.4	47.6
CP ^3^ (%)	10.5	10.5	10.5	11.3
RUP ^4^ (%CP)	31.5	30.5	30.3	27.2
RDP ^5^ (%CP)	68.5	69.6	69.7	72.8
ME ^6^ (Mcal/kg)	2.03	2.21	2.40	1.89
NE_L_ ^7^ (Mcal/kg)	1.31	1.42	1.54	1.22
ADF ^8^ (%)	34.3	29.4	24.7	36.6
NDF ^9^ (%)	52.8	45.2	37.9	57.6
Starch (%)	18.4	24.2	30.2	6.11
Sugar (%)	3.42	5.56	7.43	5.97
Ash (%)	6.42	6.08	5.67	5.96
EE ^10^ Total (%)	3.01	3.24	3.43	3.45

^1^ Dietary treatment consist of: (1) LE: Low energy, (2) ME: Medium energy, and (3) HE: High energy, LD: Lactation diet. Low, medium, and high energy levels were NE_L_: 1.30, 1.42, and 1.53 Mcal/kg, respectively, on DM basis in dietary treatments. ^2^ DM: Dry matter, ^3^ CP: Crude protein, ^4^ RUP: Ruminal undegradable protein, ^5^ RDP: Ruminal degradable protein, ^6^ ME: Metabolizable energy, ^7^ NE_L_: Net energy for lactation, ^8^ ADF: Acid detergent fiber, ^9^ NDF: Neutral detergent fiber, and ^10^ EE: Ether extract.

**Table 2 animals-12-01683-t002:** Production performance of Nili Ravi buffaloes fed different levels of energy.

	Dietary Treatments ^1^				*p*-Value ^2^		
Items	LE	ME	HE	SEM	Diet	Week	Diet × Week
DMI, kg/d	12.2	13.4	13.8	0.23	<0.01	0.06	0.57
Milk ^3^, kg/d	6.62	7.87	8.70	0.73	0.15	--	--
Milk ^4^, kg/d	7.83	8.99	9.61	0.75	0.32	<0.01	0.54
Fat ^5^, g/d	429.3	496.8	506.2	15.65	0.34	0.41	<0.01
Protein ^5^, g/d	268.3 ^a^	350.0	362.7 ^b^	32.49	0.05	0.02	0.71
Lactose ^5^, g/d	355.0 ^a^	441.0	479.5 ^b^	35.28	0.09	<0.01	0.62
Composition, %							
Fat	6.14	5.67	5.53	0.237	0.19	<0.01	0.29
Protein	3.83	3.91	3.96	0.044	0.11	<0.01	0.84
Lactose	5.20	5.30	5.37	0.056	0.12	<0.01	0.52

^a,b^ Values within a row with different superscript letters differ (*p* < 0.05). ^1^ Dietary treatment were: LE: Low energy, ME: Medium energy, and HE: High energy. Low, medium, and high energy levels were NE_L_: 1.30, 1.42, and 1.53 Mcal/kg, respectively, on DM basis in dietary treatments. ^2^ Probability of treatment effects: Treat: Main effect of treatment, week: Main effect of week, treat x week: Interaction between treatment and week effect. ^3^ Milk production across the complete lactation. ^4^ Milk production in the first 14 weeks. ^5^ Milk composition in the first 12 weeks.

**Table 3 animals-12-01683-t003:** Body condition score and body weight of Nili Ravi buffaloes fed different levels of energy.

	Dietary Treatments ^1^				*p*-Value ^2^		
Items	LE	ME	HE	SEM	Diet	Week	Diet × Week
Body condition score							
Prepartum	3.7	3.7	3.9	0.16	0.76	<0.01	<0.01
Postpartum	3.2	3.4	3.8	0.16	0.19	<0.01	0.02
Body weight, kg							
Prepartum	620.8	619.9	679.6	22.97	0.12	<0.01	0.88
Postpartum	555.7	543.9	603.4	20.93	0.13	<0.01	0.78

^a,b^ Values within a row with different superscript letters differ (*p* < 0.05). ^1^ Dietary treatment were: (1) LE: Low energy, (2) ME: Medium energy, and (3) HE: High energy. Low, medium, and high energy levels were NE_L_: 1.30, 1.42, and 1.53 Mcal/kg, respectively, on DM basis in dietary treatments. ^2^ Probability of treatment effects: Treat: Main effect of treatment, week: Main effect of week, treat × week: Interaction between treatment and week effect.

**Table 4 animals-12-01683-t004:** Serum metabolites (mg/dL) of Nili Ravi buffaloes fed different levels of energy.

	Dietary Treatments ^1^				*p*-Value ^2^		
Items	LE	ME	HE	SEM	Diet	Week	Diet × Week
Glucose							
Prepartum	75.8	79.9	71.7	6.19	0.67	0.48	0.27
Postpartum	77.1	77.2	57.6	6.45	0.10	0.33	0.87
Triglyceride							
Prepartum	177.7	144.7	117.6	33.38	0.48	0.40	0.40
Postpartum	209.6 ^a^	195.7 ^a^	117.2 ^b^	14.29	<0.05	0.13	0.56
BUN ^3^							
Prepartum	34.0	26.5	29.0	2.80	0.23	0.09	0.13
Postpartum	25.6	35.4	22.3	4.54	0.16	0.33	0.83
BHBA ^4^							
Prepartum	6.6	6.1	5.1	1.09	0.65	0.44	0.83
Postpartum	7.4	8.0	5.1	1.01	0.13	0.08	0.02

^a,b^ Values within a row with different superscript letters differ (*p* < 0.05). ^1^ Dietary treatment were: (1) LE: Low energy, (2) ME: Medium energy, and (3) HE: High energy. Low, medium, and high energy levels were NE_L_: 1.30, 1.42, and 1.53 Mcal/kg, respectively, on DM basis in dietary treatments. ^2^ Probability of treatment effects: Treat: Main effect of treatment, week: Main effect of week, treat × week: Interaction between treatment and week effect. ^3^ BUN: Blood urea nitrogen. ^4^ BHBA: β-hydroxybutyrate.

**Table 5 animals-12-01683-t005:** Postpartum reproductive performance of Nili Ravi buffaloes fed different levels of energy.

	Dietary Treatments ^1^				*p*-Value ^2^ (Diet)
Variables	LE	ME	HE	SEM	
FFF ^3^	22.2	16.5	17.2	3.64	0.49
OR ^4^	22.4	16.8	15.5	3.67	0.39
FCL ^5^	31.4	31.8	23.0	4.42	0.30
UI ^6^	61.1 ^a^	54.1 ^a,b^	43.5 ^b^	4.77	0.05
First estrus	176.0	118.2	102.2	0.00	<0.01
Days open	265.6 ^a^	183.2 ^a,b^	122.8 ^b^	35.63	0.06
Calving interval	567.6	492.4	433.7	--	<0.01
Conceived	5	7	7	--	--

^a,b^ Values within a row with different superscript letters differ (*p* < 0.05). ^1^ Dietary treatment were: (1) LE: Low energy, (2) ME: Medium energy, and (3) HE: High energy. Low, medium, and high energy levels were NE_L_: 1.30, 1.42, and 1.53 Mcal/kg, respectively, on DM basis in dietary treatments. ^2^ Probability of treatment effects: Treat: Main effect of treatment. ^3^ FFF: First follicular formation, ^4^ OR: Ovarian resumption, ^5^ FCL: First corpus luteum formation, ^6^ UI: Uterine involution.

## Data Availability

Supporting data can be provided by the first and corresponding author.

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
