# Peer review of "Effect of Prepartum Dietary Energy Level on Production and Reproduction in Nili Ravi Buffaloes"

_animals, 2022, doi:10.3390/ani12131683_

Round 1

Reviewer 1 Report

Line and Comment

76 Did you estimate the number of cows that would be needed to detect a 0.5 to 1 kg/day difference in milk yield per cow among each of the three prepartum treatments? This is an important issue for this experiment. The milk yield patterns in Figure 2 show a strong tendency for treatment differences, but the number of cows per treatment may have been inadequate to detect such differences.

83 What was the total amount of feed offered daily during the dry period?

132 performed instead of performed

133 Milk

References: The number of references should be reduced.

Using the term "opposite" in the discussion is not informative. The results may differ or be in contrast to other findings, but not clearly opposite. 

Author Response

76 Did you estimate the number of cows that would be needed to detect a 0.5 to 1 kg/day difference in milk yield per cow among each of the three prepartum treatments? This is an important issue for this experiment. The milk yield patterns in Figure 2 show a strong tendency for treatment differences, but the number of cows per treatment may have been inadequate to detect such differences.

Ans. We used the minimum numbers in the current experiment. Ideally it should be between 9-12, however, we were limited by number of animals because of parity x days remaining in calf etc… Moreover, if we review the literature of buffalo when starch is increased, it tends to modify more milk constituents compared to milk volume. We are also doing protein x fat x starch studies in lactating buffaloes and in our starch studies we don’t observe any significant changes in milk volume.

83 What was the total amount of feed offered daily during the dry period?

 Ans. Feeding was done on basis of total mixed ration (TMR) which was prepared by mixing concentrate, wheat straw and corn silage. On as such basis in low energy group, 18 kg corn silage, 5.5 kg wheat straw and 4.1 kg low energy concentrate was offered to animals whereas, in medium energy group 21 kg corn silage, 4 kg wheat straw and 4.8 kg medium energy concentrate and in high energy group 22 kg corn silage, 2.5 kg wheat straw and 6.4 kg high energy concentrate fed to dry period. On dry matter basis 15.00 kg TMR was offered in all treatment groups. The prepartum DMI has been included in Table No 2.

132 performed instead of performed

Ans. Corrected please

133 Milk

Ans Corrected please

References: The number of references should be reduced.

Ans 10 references has been reduced from 82 to 72.

Using the term "opposite" in the discussion is not informative. The results may differ or be in contrast to other findings, but not clearly opposite

Ans All "opposite" has been replaced please with word differ or contrast, please

Reviewer 2 Report

GENERAL REMARKS

Dear authors,

I have evaluated the manuscript identified as animals-1737960. I find the research addressed extremely interesting, above all due to the scarcity of useful information for the definition of nutritional standards in the dairy buffalo and, even more, about the feeding in the peripartum. Therefore, I believe that the manuscript's contribution to literature is highly appreciated. Nevertheless, in evaluating the manuscript I had to point out some inaccuracies, as well as some doubts that arose in different parts of the manuscript.

My suggestions are listed below. I hope my contribution will be useful to improve the manuscript.

Good luck and good work.

Regards

SPECIFIC COMMENTS

L12-18: in my opinion, the sentence lacks the subject. Anyway, consistent with the template of the journal, authors are encouraged to reword the simple summary as a whole. The latter, in fact, is designed to reach a wide lay audience with respect to the subject. Excessive technicalities and abbreviations should be avoided. Having to provide concise information to an uneducated audience, the simple summary should be a clear statement of the problem addressed, the aims and objectives, relevant results, conclusions from the study, and how they will be valuable to society. In conclusion, authors should avoid summarizing the abstract in a simple summary. Thanks

L22 (and along with the text): in my opinion, rewording "during " with "for" will make the meaning clearer. Thanks

L32-33 (and along with the text): authors are invited to double-check if they are sticking to singular or plural. Thanks

L39: only one punctuation mark is needed. Thanks

L45: the authors refer to the working attitude of the swamp buffalo as a fact. In my opinion, the sentence should be bibliographically supported. In this regard, I would like to bring to the attention of the authors a recent paper focused on the systematic review of the use of the buffalo as a draft and work animal (https://doi.org/10.3390/ani11092683). In my opinion, the manuscript could represent a valid tool for supporting and implementing the text parts focused on the description of the non-dairy uses of the water buffalo Thanks

L46-58: in my opinion, the authors chase the same concept in multiple sentences, making the text redundant. I apologize for pedanticism, but I believe that it is appropriate to reformulate the text in favor of greater clarity. Thanks

L51 (and along with the text): please, add a space before the square bracket. Thanks

L52 (and along with the text): please, check the space among the text's words. Thanks

L52 (and along with the text): the abbreviations should be specified at first mention. Thanks

L54 (and along with the text): according to the journal template, multiple references should be included in a single pair of parentheses. Thanks

L61-64: I can agree with the lack of in-depth knowledge about the nutritional requirements of the dairy buffalo but, at the same time, I cannot agree with the fact that under the usual farming conditions the buffaloes are routinely fed with poor quality feeds. What the authors said could be probably true in certain breeding areas or relative to specific physiological moments, but certainly not for lactating animals rearing in the Mediterranean area, where supplementation with adequate quantities of concentrates is ordinary practice. So, I suggest recasting the statements, providing a more realistic framework of the current buffalo feeding issue. In this regard, authors are encouraged to consider the manuscript https://doi.org/10.3390/ani10030515, which highlight some relevant critical issue in feeding dairy buffaloes. Thanks

L76: please, double-check the punctuation on this date. Thanks

L77 (and along with the text): although references should be numerically listed, the author's first name should be reported in the text (e.g., Edmonson et al. [24]). Thanks

L77-82: it would be desirable that the authors provide more information about the nutritional standards used both in the pre-and post-partum phases. Thanks

L85: it is not clear to me which mode of administration of the diets was used. Looking at Table 1, I can assume that the experimental diets were provided as TMR. Thanks

L86: perhaps it is appropriate, even if only as a hint, to define the weaning schedule and the diet provided to calves. Thanks

L88: LD acronymous is not mentioned in the table footnotes

L94: with reference to residues, the adjective "individual" refers to the single animal (which could be in reference to pre-parturition) or to the single ingredient (which, especially in group feeding or TMR diets, appears unlikely). Authors are asked to clarify. Thanks

L100-104: failure to milk ejection due to neurophysiological bonding between mother and calf is a possible drawback in the dairy buffalo, usually resolved with the exogenous oxytocin injection. I agree with the fact that with the sight of the calf the mothers may have released milk; nevertheless, I find the milking of the udder parts in an alternating manner unusual. The denial of milk by the buffaloes I do not think happens for an udder part. Could the authors be clearer on this? Thanks

L108: please, decline hours in the plural. Thanks

L 132: please, rewrite “perofrmed”. Thanks

L138-139: please, rewrite “varaibles” and “uterine incolution”. Thanks

L140-141 (and along with the text): according to the template of the journal, the p-value should be written in lower case and italic. Thanks

L171-172 (table 2): in the materials and methods the authors did not claim to measure milk production throughout the whole lactation. It would be appropriate to recall this. In addition, it is not clear to me why the milky composition refers to only 12 weeks. Thanks

L195, 200, 211, 218: Please, add a comma after “medium”. Thanks

L215: please, rewrite beta-hydroxy “byutyrate”. Thanks

L238-379 (discussion): I find that the focus of the discussions is very well convergent with the results but that the discussions are too much aimed at a comparative analysis with other studies rather than at a concrete argumentation of the same. For example, the phrase "our results are in agreement" occurs too frequently in discussions. In my opinion, authors should try to rearrange the discussions, focusing more on the content than on a mere and redundant comparison of their results with the research of others. Authors are also advised to take into account the recommendations made for line 77. Thanks

L250: please, BCS should be written in uppercase. Thanks

L250: please, double-check the sequence of verbs in “was might be”. Thanks

L263: about “These findings are in agreement”, please try using a verb instead of a noun phrase to be concise. Thanks

L287: a space should follow most punctuation. Thanks

L308: please, add a comma between clauses. Thanks

L315: in my opinion, "calves weight" would be better as possessive. Thanks

L356, 263: please, check the space after punctuation. Thanks

L358: see suggestion reported for L250. Thanks

L374: be careful of the singular or plural (e.g., are). Thanks

Author Response

SPECIFIC COMMENTS

L12-18: in my opinion, the sentence lacks the subject. Anyway, consistent with the template of the journal, authors are encouraged to reword the simple summary as a whole. The latter, in fact, is designed to reach a wide lay audience with respect to the subject. Excessive technicalities and abbreviations should be avoided. Having to provide concise information to an uneducated audience, the simple summary should be a clear statement of the problem addressed, the aims and objectives, relevant results, conclusions from the study, and how they will be valuable to society. In conclusion, authors should avoid summarizing the abstract in a simple summary. Thanks

Ans corrected as guided please

L22 (and along with the text): in my opinion, rewording "during " with "for" will make the meaning clearer. Thanks

Ans corrected as per suggestions please

L32-33 (and along with the text): authors are invited to double-check if they are sticking to singular or plural. Thanks

Ans corrected please

L39: only one punctuation mark is needed. Thanks

Ans corrected please

L45: the authors refer to the working attitude of the swamp buffalo as a fact. In my opinion, the sentence should be bibliographically supported. In this regard, I would like to bring to the attention of the authors a recent paper focused on the systematic review of the use of the buffalo as a draft and work animal (https://doi.org/10.3390/ani11092683). In my opinion, the manuscript could represent a valid tool for supporting and implementing the text parts focused on the description of the non-dairy uses of the water buffalo Thanks

Ans added please

L46-58: in my opinion, the authors chase the same concept in multiple sentences, making the text redundant. I apologize for pedanticism, but I believe that it is appropriate to reformulate the text in favor of greater clarity. Thanks

Ans Changes are made as guided please

L51 (and along with the text): please, add a space before the square bracket. Thanks

Ans corrected please

L52 (and along with the text): please, check the space among the text's words. Thanks

Ans corrected please

L52 (and along with the text): the abbreviations should be specified at first mention. Thanks

Ans corrected please

L54 (and along with the text): according to the journal template, multiple references should be included in a single pair of parentheses. Thanks

Ans corrected please

L61-64: I can agree with the lack of in-depth knowledge about the nutritional requirements of the dairy buffalo but, at the same time, I cannot agree with the fact that under the usual farming conditions the buffaloes are routinely fed with poor quality feeds. What the authors said could be probably true in certain breeding areas or relative to specific physiological moments, but certainly not for lactating animals rearing in the Mediterranean area, where supplementation with adequate quantities of concentrates is ordinary practice. So, I suggest recasting the statements, providing a more realistic framework of the current buffalo feeding issue. In this regard, authors are encouraged to consider the manuscript https://doi.org/10.3390/ani10030515, which highlight some relevant critical issue in feeding dairy buffaloes. Thanks

Ans. Added please

L76: please, double-check the punctuation on this date. Thanks

Ans corrected please

L77 (and along with the text): although references should be numerically listed, the author's first name should be reported in the text (e.g., Edmonson et al. [24]). Thanks

Ans corrected please

L77-82: it would be desirable that the authors provide more information about the nutritional standards used both in the pre-and post-partum phases. Thanks

Ans Nutritional standards are not available for prepartum buffaloes, and it was first study on buffaloes, while postpartum feeding was conducted as per traditional system to check the effect of prepartum feeding on postpartum performance.

L85: it is not clear to me which mode of administration of the diets was used. Looking at Table 1, I can assume that the experimental diets were provided as TMR. Thanks

Ans Feeding was done on basis of total mixed ration (TMR) which was prepared by mixing the concentrate, wheat straw and corn silage in different ratios.

L86: perhaps it is appropriate, even if only as a hint, to define the weaning schedule and the diet provided to calves. Thanks

Ans corrected please

L88: LD acronymous is not mentioned in the table footnotes

Ans corrected please

L94: with reference to residues, the adjective "individual" refers to the single animal (which could be in reference to pre-parturition) or to the single ingredient (which, especially in group feeding or TMR diets, appears unlikely). Authors are asked to clarify. Thanks

Ans corrected please

L100-104: failure to milk ejection due to neurophysiological bonding between mother and calf is a possible drawback in the dairy buffalo, usually resolved with the exogenous oxytocin injection. I agree with the fact that with the sight of the calf the mothers may have released milk; nevertheless, I find the milking of the udder parts in an alternating manner unusual. The denial of milk by the buffaloes I do not think happens for an udder part. Could the authors be clearer on this? Thanks

Ans Milking was done by milk man one side and offering one side of udder to calf as per allowance and no denial of milk was happens by buffaloes and complete milking without calf suckling was also practiced as to overcome any error and it was similar without any error.

L108: please, decline hours in the plural. Thanks

Ans corrected please

L 132: please, rewrite “perofrmed”. Thanks

Ans corrected please

L138-139: please, rewrite “varaibles” and “uterine incolution”. Thanks

Ans corrected please

L140-141 (and along with the text): according to the template of the journal, the p-value should be written in lower case and italic. Thanks

Ans corrected please

L171-172 (table 2): in the materials and methods the authors did not claim to measure milk production throughout the whole lactation. It would be appropriate to recall this. In addition, it is not clear to me why the milky composition refers to only 12 weeks. Thanks

Ans added on line number 103 and as first 12 weeks are very important for negative energy balance and any change in milk composition, so it was recorded till 12 months.

L195, 200, 211, 218: Please, add a comma after “medium”. Thanks

Ans corrected please

L215: please, rewrite beta-hydroxy “byutyrate”. Thanks

Ans corrected please

L238-379 (discussion): I find that the focus of the discussions is very well convergent with the results but that the discussions are too much aimed at a comparative analysis with other studies rather than at a concrete argumentation of the same. For example, the phrase "our results are in agreement" occurs too frequently in discussions. In my opinion, authors should try to rearrange the discussions, focusing more on the content than on a mere and redundant comparison of their results with the research of others. Authors are also advised to take into account the recommendations made for line 77. Thanks

Ans"our results are in agreement" have been changed please

L250: please, BCS should be written in uppercase. Thanks

Ans Corrected please

L250: please, double-check the sequence of verbs in “was might be”. Thanks

Ans corrected please

L263: about “These findings are in agreement”, please try using a verb instead of a noun phrase to be concise. Thanks

Ans corrected please

L287: a space should follow most punctuation. Thanks

Ans corrected please

L308: please, add a comma between clauses. Thanks

Ans corrected please

L315: in my opinion, "calves weight" would be better as possessive. Thanks

Ans corrected please

L356, 263: please, check the space after punctuation. Thanks

Ans corrected please

L358: see suggestion reported for L250. Thanks

Ans corrected please

L374: be careful of the singular or plural (e.g., are). Thanks

Ans corrected please

Reviewer 3 Report

This is a well written piece of work at which the effects of  the plane of nutrition during the dry period on production and fertility indices are examined. The study provides the reader with some interesting findings, but some issues are imperative to be addressed before the acceptance for publication.

1.     Three different energy levels were used. It must be clearly stated what is the recommended level (if this information is available) and the rate of surplus or deficit  

2.     The number of animals is very small to allow sound conclusions to be drawn on reproductive parameters; hence, the authors must be very careful when discussing their findings.

3.     In group LE two animals were culled due to severe emaciation. What was behind the excessive weight loss? Was there a health issue or the low energy at dry period was the causative factor. In any case, all data obtained from these two animals should be totally excluded from the analysis ( not because they filed to conceive, but because an inappropriate comparison between healthy and sick animals is highly likely).

4.     The discussion is very weak. Solely comparing the results of the study with those from other published works, does not consist a scientific discussion. The results must be interpreted on a biological basis, under the light of the existing knowledge. Further, the authors should be more careful on their interpretations attempts, and they must include to the discussion some findings that worth interpretation. For example: 

a.     The differences on milk production are discussed, while their statistics revealed no difference. Differences exist only when statistics say so.

b.      Prepartum BHBA levels in ME group tended (p=0.08) to differ from other groups despite the lack of difference in milk yield and the changes in BCS. How this could be explained?

c.     It is very simplistic to say that the more fat in LE group is related to high fat mobilization, when NEFA levels were not evaluated, and no difference existed in either milk yield, or in postpartum BCS changes between groups.

d.     In the BCS graph it appears that in LE group there is a profound reduction in the mean BCS, that is not accompanied with a significant increase in milk yield. What is the possible explanation of this?  Is this reduction related to the inclusion of the two (sick??) emaciated cows?

Author Response

  1. Three different energy levels were used. It must be clearly stated what is the recommended level (if this information is available) and the rate of surplus or deficit  
  2. Ans Recommendation level were not available for buffaloes, and these levels were selected based on NRC 2001 recommendation which suggest feeding a pregnant dry cow at a level of 1.44 Mcal/Kg.
  3. The number of animals is very small to allow sound conclusions to be drawn on reproductive parameters; hence, the authors must be very careful when discussing their findings.
  4. Ans Although numbers of animals were low but we can conclude that LE diet adversely affected all productive and reproductive parameters.
  5. In group LE two animals were culled due to severe emaciation. What was behind the excessive weight loss? Was there a health issue or the low energy at dry period was the causative factor. In any case, all data obtained from these two animals should be totally excluded from the analysis (not because they filed to conceive, but because an inappropriate comparison between healthy and sick animals is highly likely).

Ans, The BCS loss was observed among all animals of LE group, because low energy feeding during prepartum feeding adversely affected their performance, and possible low DMI during prepartum might influenced low DMI during postpartum feeding. They never remained or expressed any symptoms of sick animals, but were culled at the end of year due to normal culling process of farm as their BCS was lower than required.

  1. The discussion is very weak. Solely comparing the results of the study with those from other published works, does not consist a scientific discussion. The results must be interpreted on a biological basis, under the light of the existing knowledge. Further, the authors should be more careful on their interpretations attempts, and they must include to the discussion some findings that worth interpretation. For example: 
  2. The differences on milk production are discussed, while their statistics revealed no difference. Differences exist only when statistics say so.

Ans Milk production was higher in HE group compared to LE group but low number of animals might be possible reason that it not differ statistically.

  1. Prepartum BHBA levels in ME group tended (p=0.08) to differ from other groups despite the lack of difference in milk yield and the changes in BCS. How this could be explained?

Ans Prepartum BHBA level remained similar among all groups, while postpartum BHBA tended to lower in HE group, indicate more fat reserves for animals of HE group.

  1. It is very simplistic to say that the more fat in LE group is related to high fat mobilization, when NEFA levels were not evaluated, and no difference existed in either milk yield, or in postpartum BCS changes between groups.

Ans milk yield and BCS changes remained lowest in LE group but not differ statistically might due to low number of experimental animals in each group.

  1. In the BCS graph it appears that in LE group there is a profound reduction in the mean BCS, that is not accompanied with a significant increase in milk yield. What is the possible explanation of this?  Is this reduction related to the inclusion of the two (sick??) emaciated cows?

Ans Low energy diet feeding for 63 days exerted negative impact especially on dry matter intake and all animals start losing body weight and BCS throughout the experiment and overall milk production at the end of lactation was also lower in LE group. No animal from LE group remained sick or expressed sick symptoms till end of trial. It was might be due to the lack of reserves for milk synthesis for LE group, and milk production of all experimental animals in LE group remained lower and similar. 

Round 2

Reviewer 1 Report

 I recommend accepting this paper.

Reviewer 2 Report

Dear authors,
I have evaluated the revised version of the manuscript identified as animals-1737960. Given the changes made, I have no doubts about the manuscript's publishable status. Overall, I believe that the manuscript can actively contribute to filling the lack of information about the nutritional requirements of the dairy buffaloes, and, for this, I congratulate the authors. As a sole and final recommendation, I wanted to point out to the authors that, as per the journal' template, the reference named [1.1] should be [2]. In addition, both the names of the first and last author should be correct. Finally, both this and other references (for example reference 23 in the current version) should be accompanied by the DOI, if available.
I congratulate again
Regards

Reviewer 3 Report

Unfortunately, the authors insisted on their original version, and preferred to ignore my advise given towards the improvement of their manuscript. The minor changes made did not improve the quality of the paper. Consequently, this reviewer believes that the paper should not be given priority for publication in Animals